# Systemic Oxygen Delivery during One-Lung Ventilation: Comparison between Propofol and Sevoflurane Anaesthesia in a Randomised Controlled Trial

**DOI:** 10.3390/jcm8091438

**Published:** 2019-09-11

**Authors:** Tae Soo Hahm, Heejoon Jeong, Hyun Joo Ahn

**Affiliations:** Department of Anaesthesiology and Pain Medicine, Samsung Medical Center, Sungkyunkwan University School of Medicine, Seoul 06351, Korea; ts.hahm@samsung.com (T.S.H.); heejoon.jeong@samsung.com (H.J.)

**Keywords:** Delivery of oxygen, one-lung ventilation, propofol, sevoflurane, thoracic anaesthesia

## Abstract

Systemic oxygen delivery (DO_2_) is a more comprehensive marker of patient status than arterial oxygen saturation (SaO_2_), and DO_2_ in the range of 330–500 mL min^−1^ is reportedly adequate during anaesthesia. We measured DO_2_ during one-lung ventilation (OLV) for thoracic surgery—where the risk of pulmonary shunt is significant, and hypoxia occurs frequently—and compared sevoflurane and propofol, the two most commonly used anaesthetics in terms of DO_2_. Sevoflurane impairs hypoxic pulmonary vasoconstriction. Thus, our hypothesis was that propofol-based anaesthesia would show a higher DO_2_ value than sevoflurane-based anaesthesia. This was a double-blinded randomised controlled trial conducted at a university hospital from 2017 to 2018. The study population consisted of patients scheduled for lobectomy under OLV (*N* = 120). Sevoflurane or propofol was titrated to a bispectral index of 40–50. Haemodynamic variables were measured during two-lung ventilation (TLV) and OLV at 15 and 45 min (OLV15 and OLV45, respectively) using oesophageal Doppler monitoring. The mean DO_2_ (mL min^−1^) was not different between the sevoflurane and propofol anaesthesia groups (TLV: 680 vs. 706; OLV15: 685 vs. 703; OLV45: 759 vs. 782, respectively). SaO_2_ was not correlated with DO_2_ (r = 0.09, *p* = 0.100). Patients with SaO_2_ < 94% showed adequate DO_2_ (641 ± 203 mL min^−1^), and patients with high SaO_2_ (> 97%) showed inadequate DO_2_ (14% of measurements < 500 mL min^−1^). In conclusion, DO_2_ did not significantly differ between sevoflurane and propofol. SaO_2_ was not correlated with DO_2_ and was not informative regarding whether the patients were receiving an adequate oxygen supply. DO_2_ may provide additional information on patient status, which may be especially important when patients show a low SaO_2_.

## 1. Introduction

The concept of systemic delivery of oxygen (DO_2_) is attracting increasing interest in both the context of anaesthesia [1] and the intensive care unit (ICU) [2,3,4]. DO_2_ is calculated as: (haemoglobin × 1.34 × SaO_2_ + PaO_2_ × 0.003) × cardiac output × 10. It is, thus, a more comprehensive (and important) marker of patient status than arterial oxygen saturation (SaO_2_) [5], being based on haemoglobin, oxygen saturation, and cardiac output.

Normal DO_2_ in awake, healthy, subjects is 1000 mL min^−1^ at rest, while O_2_ consumption (VO_2_) is 250 mL min^−1^ [5]. A target DO_2_ of 330 [6] or 500 mL min^−1^ [7,8] has been suggested for preventing tissue O_2_ deprivation under anaesthesia.

There is a significant risk of pulmonary shunt and hypoxia during one-lung ventilation (OLV) in thoracic surgery. Sevoflurane and propofol, the two most common anaesthetics, have been compared in terms of SaO_2_, but not in terms of DO_2_, during OLV [9].

These anaesthetics may show different associations with DO_2_, based on their differential effects on SaO_2_ and cardiac output (which are the major determinants of DO_2_). Inhalation anaesthetics, including sevoflurane, are thought to reduce hypoxic pulmonary vasoconstriction [10,11], thereby increasing the ‘shunting’ of nonoxygenated blood during OLV and, thus, causing lower SaO_2_ [12]. Notably, lower SaO_2_ can decrease DO_2_. It is not clear whether sevoflurane or propofol yields a higher cardiac output [13,14,15,16]. Therefore, in the present study, we measured DO_2_ in patients undergoing thoracic surgery with OLV and compared differences therein between sevoflurane- and propofol-based anaesthesia. We hypothesised that sevoflurane-based anaesthesia would be associated with a lower DO_2_ than propofol-based anaesthesia. The secondary outcome was the relationship between SaO_2_ and DO_2_.

## 2. Experimental Section

This prospective, randomised study was approved by the Institutional Review Board of Samsung Medical Center, Chairperson Prof. Suk-Koo Lee, Seoul, Korea (IRB file number: SMC 2017-06-069-003, IRB approval: 2017-09-05) and registered prior to patient enrolment at the Clinical Research Information Service (KCT0002782; Principal investigator, Tae Soo Hahm; date of first submission, 20 September 2017; date of registration of first patient, 25 September 2017; https://cris.nih.go.kr/cris). Written informed consent was obtained from all participants.

### 2.1. Study Population

This study was performed between September 2017 and July 2018 at the Samsung Medical Center (Seoul, Korea). During the study period, a total of 144 patients were assessed for eligibility by study staff, and 139 patients were enrolled in the study.

The inclusion criteria were age ≥ 19 years, American Society of Anaesthesiologists physical status I-III, and elective pulmonary lobectomy under open thoracotomy or video-assisted thoracoscopic surgery (VATS). Cases requiring at least 1 h of OLV were included. The exclusion criteria were forced expiratory volume in 1 s (FEV_1_) < 40% of predicted value, cardiac ejection fraction < 50%, recent oesophageal surgery or presence of congenital or acquired oesophageal abnormalities (stricture, varices, or fistula), and haemoglobin < 10 g dL^−1^. Patients who did not understand the study objectives or refused to participate were excluded. Dropout criteria included OLV < 1 h, protocol interruption for rescue ventilation (SpO_2_ < 90%), bleeding > 400 mL, sampling/measurement error, and inotrope or vasopressor administration during measurement.

### 2.2. Randomisation and Blinding Procedure

Patients were randomised into sevoflurane and propofol groups by computer-generated random numbers with a fixed block size of 4 and a 1:1 ratio, and patient allocations were sealed in an opaque envelope. An attending anaesthesiologist who was not involved in the study opened the sealed envelope just before induction of anaesthesia and provided the designated anaesthetic agents according to the group assignment. Oesophageal Doppler monitoring was performed by a single designated anaesthesiologist who was not involved in the study, while vaporiser, gas monitor, and drug infusion pumps were hidden by an opaque screen. The corresponding author and co-authors collected data by retrieving blinded study logs. Attending anaesthesiologists were not blinded to the patients’ group assignment, but they were not involved in patient allocation or data analysis.

### 2.3. Intraoperative Management

No premedication was given before induction of anaesthesia. After the patient arrived at the operation room, electrocardiography, a non-invasive blood pressure cuff, pulse oximetry, and bispectral index (BIS) monitor (v. 4.0; Aspect Medical Systems, Natick, MA, USA) were applied.

For induction of anaesthesia, a bolus of 1.5–2.5 mg kg^−1^ propofol with continuous remifentanil infusion (0.05 μg kg^−1^ min^−1^) was used. During surgery, anaesthesia was maintained with sevoflurane or propofol. Propofol was administered using an infusion pump in the range of 80–120 μg kg^−1^ min^−1^. The attending anaesthesiologist titrated sevoflurane or propofol to maintain the BIS index between 40 and 50. Remifentanil (0.05 μg kg^−1^ min^−1^) and rocuronium were continuously infused in both groups. Bolus administration of remifentanil (0.3 μg kg^−1^) was used for intubation and during intensive surgical stimulation.

Intubation was performed using a double-lumen tube after 1.0 mg kg^−1^ rocuronium bolus injection, and the position of the tube was confirmed by fibreoptic bronchoscopy. A radial arterial catheter was placed on the opposite side of surgery. After induction, the oesophageal Doppler probe (CardioQ; Deltex Medical, Irving, TX, USA) was inserted through the oropharynx into the distal oesophagus, approximately 35–50 cm from the incisors. Ringer’s solution was used as the maintenance fluid and was infused at 3–5 mL kg^−1^ h^−1^.

All patients received the same ventilation protocol, which was tidal volume of 6–8 mL kg^−1^ predicted body weight with 5 cmH_2_O of positive end-expiratory pressure during two-lung ventilation (TLV) under volume-controlled mode. Tidal volume decreased to 5–6 mL kg^−1^ predicted body weight during OLV. The ventilation rate was adjusted to maintain end-tidal carbon dioxide at 35–40 mmHg. FIO_2_ was maintained at 100% throughout the study period and was decreased to 50% thereafter. The operation was performed in the lateral decubitus position with the operated side up. OLV was started when the patient was turned to the lateral decubitus position in VATS or when the fascia was incised during open thoracotomy. All patients were extubated upon meeting the extubation criteria and transferred to the ICU after anaesthesia recovery at the post-anaesthetic care unit.

### 2.4. Measurements

Haemodynamic measurements and arterial blood gas analyses were conducted during TLV in the lateral position immediately before OLV (TLV) and at 15 and 45 min after the initiation of OLV (OLV15 and OLV45, respectively). Stable heart rate and mean arterial pressure were observed for 10 min before haemodynamic measurements without application of vasopressors/inotropes.

Doppler monitoring was performed at TLV, OLV15, and OLV45 by the designated anaesthesiologist. The oesophageal Doppler probe was manipulated with adjustment of depth and rotational position until the characteristic descending thoracic aortic waveform shape was visualised and the distinctive Doppler ‘whip crack’ sound associated with aortic blood flow was heard. The best three waveforms were stored, and the averaged values were used for determination of cardiac output.

The CardioQ and an oesophageal Doppler monitor system continuously monitored stroke volume and cardiac output without external calibration. Stroke volume was calculated as the product of the velocity-time integral, and a calibration factor was derived from a nomogram based on each patient’s age, height, and weight. DO_2_ was calculated as: (haemoglobin × 1.34 × SaO_2_ + PaO_2_ × 0.003) × cardiac output × 10. Alveolar O_2_ pressure (PAO_2_) was calculated under the high FIO_2_ condition as: FIO2 × (P_b_ − P_H2O_) − PACO_2_ = FIO_2_ × (760 − 47) − PACO_2_.

### 2.5. Statistical Analysis

The primary endpoint was the difference in DO_2_ between the two anaesthetic groups. The secondary endpoint was the correlation between SaO_2_ and DO_2_. Power analysis showed that a difference in DO_2_ of 150 mL min^−1^ between the sevoflurane and propofol groups could be regarded as significant. The standard deviation (SD) of each anaesthetic was obtained from previous studies conducted during the induction periods (sevoflurane, 72 mL min^−1^; propofol, 336 mL min^−1^). Assuming a similar reduction and a 20% dropout rate, 120 patients were required for a two-sided alpha of 5% with 80% power (independent *t* test).

Continuous variables are presented as the mean ± SD or median (interquartile range). Categorical variables are presented as counts (%). The DO_2_, cardiac output, PaO_2_/FIO_2_, alveolar arterial O_2_, and blood lactate were compared between the two groups using the independent *t* test or Mann-Whitney test depending on the data distribution. Comparisons between variables at each time point were performed using repeated-measures ANOVA. The Bonferroni correction was performed for multiple testing. The normality of the distribution of the data was evaluated with the Shapiro-Wilk test. Confidence intervals for non-normally distributed variables were calculated using the Hodges-Lehmann estimator. All *p*-values were two-sided, and *p* < 0.05 was taken to indicate statistical significance. Data were analysed using MedCalc for Windows (ver. 7.3; MedCalc Software, Mariakerke, Belgium) and SPSS software (ver. 25.0; IBM Corp., Chicago, IL, USA).

## 3. Results

A total of 144 patients were assessed for eligibility. Four patients refused to participate, and one surgery was cancelled; therefore, 139 patients were enrolled in the study. Six patients dropped out because of measurement error or missing data. One patient was converted to open-and-close surgery. OLV did not last 1 h in two patients. Ten patients required intervention for rescue ventilation due to hypoxia (SpO_2_ < 90%). There were no cases with significant intraoperative blood loss (>400 mL). No vasopressors/inotropes were administered during the haemodynamic measurement. Finally, 60 patients in each group (sevoflurane and propofol) were included in the analysis (Figure 1).

In comparisons between the two anaesthesia groups, there were no differences in baseline demographic or operational characteristics (Table 1)

DO_2_ was not different between the sevoflurane and propofol groups (TLV: 680 vs. 706 mL min^−1^, respectively; OLV15: 685 vs. 703 mL min^−1^, respectively; OLV45: 759 vs. 782 mL min^−1^, respectively; all, *p* > 0.05) (Table 2, Figure 2) and increased with time. There was no difference in SaO_2_ between the sevoflurane and propofol groups (TLV: 98.8% vs. 98.8%, respectively; OLV15: 97.7% vs. 97.8%, respectively; OLV45: 97.4% vs. 97.8%, respectively; all, *p* > 0.05) (Table 2).

Stroke volume was higher in the propofol group (TLV: 54 vs. 60 mL, *p* = 0.037; OLV15: 61 vs. 63 mL, *p* = 0.507; OLV45: 62 vs. 70 mL, *p* = 0.072, for sevoflurane vs. propofol, respectively, Bonferroni correction). Heart rate and cardiac output (TLV: 3.8 vs. 3.9 L min^−1^; OLV15: 4.0 vs. 4.0 L min^−1^; OLV45: 4.4 vs. 4.5 L min^−1^, for sevoflurane vs. propofol, respectively) were not different between the two groups (Table 2).

The alveolar-arterial O_2_ difference, which reflects pulmonary shunt, was not different between the two groups (TLV: 170 vs. 179 mmHg; OLV15: 413 vs. 405 mmHg; OLV45: 390 vs. 387 mmHg, for sevoflurane vs. propofol, respectively). PaO_2_/FIO_2_ was not different between the two groups (TLV: 483 vs. 479 mmHg; OLV15: 235 vs. 249 mmHg; OLV45: 244 vs. 257 mmHg, for sevoflurane vs. propofol, respectively).

The plasma lactate level was higher in the sevoflurane group than the propofol group (TLV: 1.39 vs. 1.23 mmol L^−1^, *p* = 0.063; OLV15: 1.40 vs. 1.23 mmol L^−1^, *p* = 0.051; OLV45: 1.42 vs. 1.21 mmol L^−1^, *P*=0.006; for sevoflurane vs. propofol, respectively, Bonferroni correction) (Table 2). DO_2_ was not correlated with SaO_2_ (r = 0.09, *p* = 0.100, Figure 3; sevoflurane group, r = 0.02; propofol group, r = 0.16).

The DO_2_ cut-off for the lowest 10th percentile was 478 mL min^−1^ (36 of 360 measurements), while the mean SaO_2_ was 97.5% for the lowest 10th percentile. Using a DO_2_ cut-off of 500 mL min^−1^, in accordance with previous studies [7,8], 52 of 360 measurements (14%) were lower than 500 mL min^−1^, and the mean SaO_2_ was 97.6% for those measurements. The lowest DO_2_ was 255 mL min^−1^, and the SaO_2_ was 98.7% at that point. DO_2_ was well-maintained at 641 mL min^−1^ in patients with SaO_2_ < 94% (21 of 360 measurements) (Table 3).

## 4. Discussion

In this study, we found no difference in DO_2_ between sevoflurane- and propofol-based anaesthesia. Furthermore, SaO_2_ was not correlated with, and did not reflect the level of, DO_2_.

Sevoflurane and propofol, the two most commonly used anaesthetics, have previously been compared in terms of SaO_2_, but not in terms of DO_2_, during OLV [9]. DO_2_ reflects the circulation and oxygenation status, and it is increasingly being used in critical care [4]. Our study is the first to measure DO_2_ during OLV and to investigate whether DO_2_ differs according to the anaesthetic used.

Previously, inhalation anaesthetics, including sevoflurane, were thought to reduce hypoxic pulmonary vasoconstriction [10,11], thereby increasing the ‘shunting’ of nonoxygenated blood during OLV and, thus, causing lower SaO_2_ [12]. In the current study, there was no difference between the two anaesthetic groups in the alveolar-arterial O_2_, which reflects pulmonary shunting. Therefore, our results are consistent with the findings of previous reports, which suggested that sevoflurane and propofol had similar effects on shunt fraction [17] and SaO_2_ [9] during OLV.

Cardiac output was not different between the two groups in this study. Previous studies reported inconsistent results regarding cardiac output in association with sevoflurane and propofol [14,15,16,18]. One study showed that when the anaesthetic was titrated to ~1 minimum alveolar concentration (MAC) to maintain the BIS between 40 and 60, the cardiac-suppressive effect was negligible [16]. Cardiac output subsequently increased during OLV in both groups in our study. Therefore, the increase in DO_2_ with time seems to be due to the increase in cardiac output over time.

The lack of difference in DO_2_ between our groups may be explained by the lack of difference in shunt amount and cardiac output between the two anaesthetics at 1 MAC and BIS 40–60.

In a normal 75 kg adult partaking in low-intensity daily activities, the amount of O_2_ consumption (VO_2_) is approximately 250 mL min^−1^ [5]. During anaesthesia, VO_2_ was reported to be 175 mL min^−1^ in one study [19]. However, VO_2_ varies considerably among patients, as well as over time, during anaesthesia [7,8]. In a recent meta-analysis, VO_2_ was shown to decrease (by −65 mL min^−1^) from baseline following the induction of anaesthesia. Moreover, it increased after surgical incision and during the postoperative period [20]. Shibutani et al. [6] suggested that a DO_2_ of at least 330 mL min^−1^ is required under anaesthesia to prevent tissue O_2_ deprivation. Skykes et al. [7,8] suggested that anaesthetists using low flows should aim for a DO_2_ closer to 500 mL min^−1^, which should also be the target in emergency situations. In high-risk patients undergoing major noncardiac surgery, the critical threshold for DO_2_ was reported as 390 mL min^−1^ m^2−1^ during anaesthesia [21]. We found that DO_2_ was generally maintained above 500 mL min^−1^ during OLV (693, 694, and 770 mL min^−1^ at TLV, OLV15, and OLV45, respectively).

There was no correlation between SaO_2_ and DO_2_ (r = 0.09, *p* = 0.100, Figure 3). DO_2_ was maintained at 641 mL min^−1^ in patients with SaO_2_ < 94%. Importantly, a high SaO_2_ does not guarantee that a patient is receiving adequate DO_2_; 14% of our DO_2_ measurements were below 500 mL min^−1^, which was regarded as the safety cut-off in previous studies [7,8]. The mean SaO_2_ was 97.6% for those measurements. The mean SaO_2_ was 97.5% in the lowest 10th percentile of DO_2_ (cut-off: 412 mL min^−1^). Based on the maximum oxygen extraction ratio (70%), patients in the lowest 10th percentile of DO_2_ are in the “danger zone” [5]. In our study, the SaO_2_ was 98.7% at the lowest DO_2_ (255 mL min^−1^).

If accompanied by high DO_2_, low SaO_2_ usually arises from increased cardiac output and subsequently increased pulmonary shunt during thoracic surgery. Therefore, a low SaO_2_ does not result in inadequate oxygen delivery if the cardiac output and DO_2_ are well maintained [5]. Our results support the necessity of measurement of DO_2_ during OLV.

In this study, the plasma lactate level was higher with sevoflurane than with propofol. The aetiology of this difference is unclear, but it may have been related to the tendency towards a lower stroke volume in the sevoflurane group than in the propofol group. Lactate is a metabolite associated with inadequate DO_2_ to tissues and is, therefore, widely used as a surrogate for tissue hypoxia. DO_2_ reflects haemodynamics and real-time systemic oxygen delivery. Therefore, DO_2_ and lactate can be used to complement each other.

This study had several limitations. Firstly, we used oesophageal Doppler monitoring to measure cardiac output instead of the thermodilution technique. However, a pulmonary artery catheter is rarely used for thoracic surgery, while oesophageal Doppler monitoring has high validity for determining changes and trends in cardiac output and is closely correlated with pulmonary artery catheter and echocardiography data [22,23,24]. However, DO_2_ is calculated based on cardiac output values on oesophageal Doppler monitoring, and uncertainties in the cardiac output measurements may have influenced the results. Secondly, DO_2_ represents global oxygen delivery, and not tissue oxygen delivery specifically. However, if patients do not have microcirculation or cellular oxygen uptake abnormalities, which are typically observed in severe vascular disease or sepsis, DO_2_ closely reflects tissue oxygen delivery. Thirdly, we excluded patients with severe pulmonary or cardiovascular dysfunction. Haemodynamic suppression may be more severe, and pulmonary shunt may show a greater increase by anaesthesia in these patients; the effect may be different between sevoflurane and propofol. This study is the first to measure DO_2_ change during OLV in relatively healthy patients. Based on our results, future studies with more seriously ill patients should be possible.

## 5. Conclusions

In this study, the type of anaesthetic (propofol or sevoflurane) did not have a significant impact on DO_2_. Furthermore, we found no correlation between SaO_2_ and DO_2_. DO_2_ data may provide useful additional information on patient status, especially in those with a low SaO_2_ level.

## Figures and Tables

**Figure 1 jcm-08-01438-f001:**
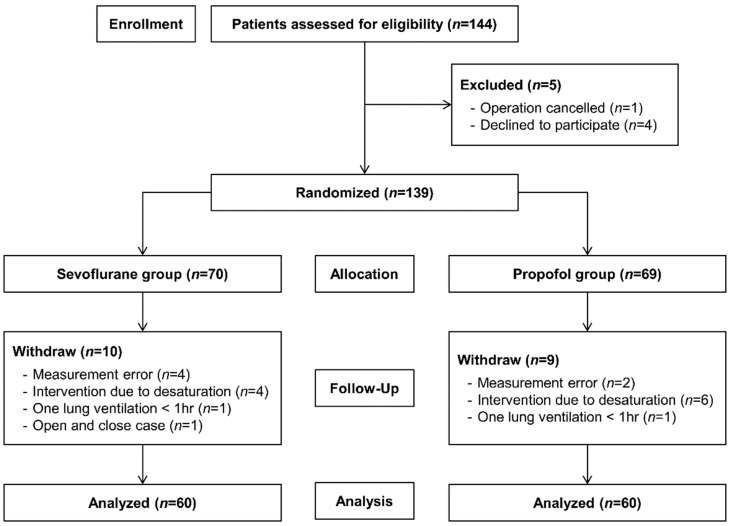
Flow diagram of patient selection.

**Figure 2 jcm-08-01438-f002:**
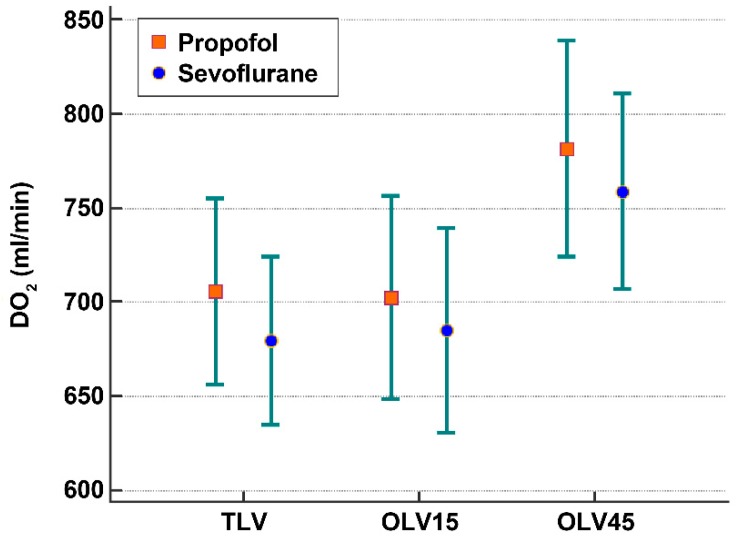
DO_2_ between sevoflurane and propofol in each time point. There was no difference in DO_2_ between the propofol and sevoflurane groups. DO_2_ increased with time. *p* = 0.0001 between TLV and OLV45, *p* = 0.0001 between OLV15 and OLV45, Bonferroni correction. TLV, two lung ventilation; OLV15, 15 min after initiation of one-lung ventilation; OLV45, 45 min after initiation of one-lung ventilation.

**Figure 3 jcm-08-01438-f003:**
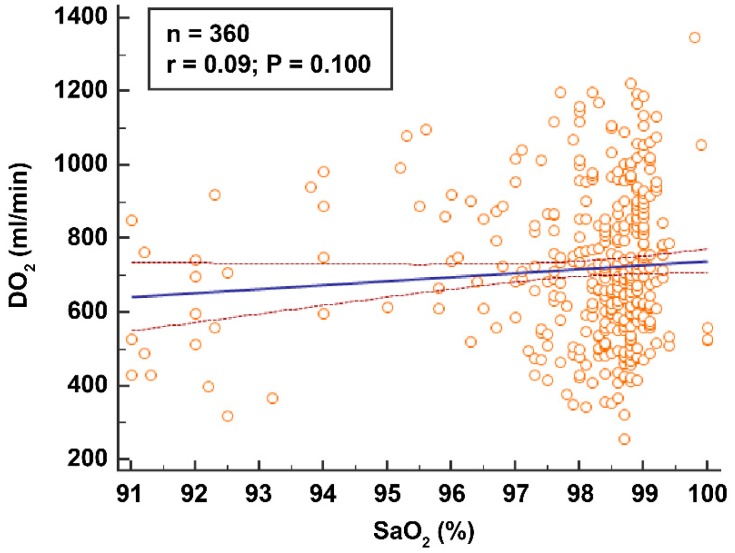
DO_2_ was not correlated with SaO_2_ (r = 0.09, *p* = 0.100). Lines are a regression line with 95% confidence intervals. A dot is each measurement (*n* = 360). TLV, two lung ventilation; OLV15, 15 min after initiation of one-lung ventilation; OLV45, 45 min after initiation of one-lung ventilation.

**Table 1 jcm-08-01438-t001:** Characteristics of patients between sevoflurane and propofol groups.

Characteristics	Sevoflurane Group (*n* = 60)	Propofol Group (*n* = 60)
Age (year)	64 (57–71)	62 (55–68)
Male	37 (61.7)	36 (60.0)
Body mass index (kg/m^2^)	24 (22–26)	24 (21–25)
ASA physical status, 1/2/3	29/23/8	25/28/7
Haemoglobin (g/dL)	13.5 (12.4–14.4)	13.2 (12.5–15.0)
Albumin (g/dL)	4.5 (4.2–4.8)	4.5 (4.2–4.7)
History of previous lung surgery	3 (5)	2 (3)
Smoking	7 (12)	3 (5)
Hypertension	18 (30)	16 (27)
Diabetes mellitus	11 (18)	6 (10)
Pulmonary comorbidities		
Recent respiratory infection	3 (5)	1 (2)
History of pulmonary tuberculosis	1 (2)	2 (3)
COPD	2 (3)	2 (3)
Bronchiectasis	0 (0)	3 (5)
Cardiac disease	3 (5)	4 (7)
Liver disease	7 (12)	9 (15)
Renal disease	4 (7)	4 (7)
Previous chemotherapy and radiotherapy	5 (8)	4 (7)
Surgery, Open/VATS	12/48	13/47
Ventilation site, Left/Right	40/20	46/14
Duration of surgery (min)	126 (93–157)	124 (100–158)
Anaesthesia time (min)	176 (142–225)	176 (147–202)
Duration of one lung ventilation (min)	100 (73–146)	101 (78–128)
Intraoperative fluid amount (mL)	900 (650–1150)	925 (750–1150)
Intraoperative blood loss (mL)	100 (50–187)	100 (50–150)
Bispectral index	45 ± 3	44 ± 2

The data are presented as mean ± standard deviation, median (interquartile range), or number (percentage). History of previous lung surgery included any kind of operation that invaded the pleural space. Smoking was defined as patients who kept smoking or stopped smoking within 1 month before surgery. Recent respiratory infection was defined as pulmonary infection within 1 month from surgery. Cardiac disease included any histories of angina and myocardial infarction. Renal disease was estimated with a glomerular filtration rate of <60 mL min^−1^ 1.73 m^2−1^. COPD, chronic obstructive pulmonary disease; VATS, video-assisted thoracoscopic surgery.

**Table 2 jcm-08-01438-t002:** Major haemodynamic variables.

Variables	TLV	OLV 15	OLV 45
DO_2_ (mL/min)			
Sevoflurane	680 ± 173	685 ± 209	759 ± 201
Propofol	706 ± 191	703 ± 208	782 ± 222
SaO_2_ (%)			
Sevoflurane	98.8 ± 0.5	97.7 ± 2.0	97.4 ± 2.0
Propofol	98.8 ± 0.4	97.8 ± 1.8	97.8 ± 1.7
Stroke volume (mL)			
Sevoflurane	54 ± 15 *	61 ± 23	62 ± 22
Propofol	60 ± 14	63 ± 20	70 ± 24
Heart rate (per min)			
Sevoflurane	70 ± 11	71 ± 11	75 ± 11
Propofol	68 ± 12	69 ± 12	71 ± 13
Mean arterial pressure (mmHg)			
Sevoflurane	85 ± 14	85 ± 14	81 ± 11
Propofol	87 ± 14	87 ± 14	80 ± 11
Cardiac output (L/min)			
Sevoflurane	3.8 ± 1.0	4.0 ± 1.2	4.4 ± 1.1
Propofol	3.9 ± 1.0	4.0 ± 1.0	4.5 ± 1.1
Haemoglobin (g/dL)			
Sevoflurane	12.6 ± 1.2	12.6 ± 1.1	12.6 ± 1.2
Propofol	12.7 ± 1.3	12.7 ± 1.5	12.6 ± 1.3
Alveolar-arterial O_2_ difference (mmHg)			
Sevoflurane	181 ± 92	426 ± 107	366 ± 146
Propofol	190 ± 101	418 ± 105	367 ± 121
PaO_2_/FIO_2_			
Sevoflurane	483 ± 88	235 ± 110	244 ± 133
Propofol	479 ± 105	249 ± 108	257 ± 121
Plasma lactate (mmol/L)			
Sevoflurane	1.39 ± 0.49	1.40 ± 0.53	1.42 ± 0.48 ^†^
Propofol	1.23 ± 0.39	1.23 ± 0.37	1.21 ± 0.36
Anion gap (mmol/L)			
Sevoflurane	11.4 ± 1.8	11.1 ± 2.0	11.1 ± 2.5
Propofol	11.1 ± 2.0	10.9 ± 1.9	10.3 ± 2.8

The data are presented as mean ± SD. * *p* = 0.037 and ^†^
*p* = 0.006, compared to the propofol group. Bonferroni correction. DO_2_, oxygen delivery; SaO_2_, arterial oxygen saturation; PaO_2_, partial pressure of arterial oxygen; FIO_2_, fraction of inspired oxygen; TLV, two-lung ventilation; OLV, one-lung ventilation.

**Table 3 jcm-08-01438-t003:** Relationship between DO_2_ and SaO_2_.

Categories	Mean DO_2_	Mean SaO_2_
DO_2_, at <lower 10th percentile	412 ± 52	97.5 ± 2.3
DO_2_, at <500 mL min^−1^ cut-off	435 ± 56	97.6 ± 2.2
DO_2_, at the lowest	255	98.7
DO_2_, at SaO_2_ < 94%	641 ± 203	92.4 ± 1.1

The data are presented as mean ± standard deviation.

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
