# Peer review of "Systemic Oxygen Delivery during One-Lung Ventilation: Comparison between Propofol and Sevoflurane Anaesthesia in a Randomised Controlled Trial"

_jcm, 2019, doi:10.3390/jcm8091438_

Round 1

Reviewer 1 Report

Hahm and coauthors measured systemic delivery of oxygen (DO2) in patients undergoing thoracic surgery with one-lung ventilation (OLV) and compared differences therein between sevoflurane- and propofol-based anaesthesia. They also evaluated the relationship between arterial oxygen saturation (SaO2) and DO2.

I have some comments.

What are the hypotheses of this study? What did authors want to prove in this study? Please clarify in Introduction.

Measurements of haemoglobin, oxygen saturation, and cardiac output are necessary to calculate DO2. It may be enough to deal with each abnormal value without calculating DO2. Why is it important to calculate DO2?

Why did authors compare sevoflurane and propofol for DO2? Are both anaesthetics predicted to have a different effect on DO2? Please clarify.

Remifentanil was infused at 0.05 μg kg-1 min-1 during surgery. (Page 3, Line 101-102) The dose of remifentanil was very low. It may be difficult to maintain blood pressure and heart rate with such low dose of remifentanil intraoperatively. How did authors deal with intraoperative analgesia? Was epidural anaesthesia performed in this study?

How did authors administer propofol? Did authors use target controlled infusion?

In this study, DO2 increased with time. (Figure 2) Why did it increase with time? Please discuss.

In this study, plasma lactate level was higher in the Sevoflurane group than in the Propofol group. Why was it higher in the Sevoflurane group?

Authors found that SaO2 was not correlated with DO2. Was this phenomenon applicable to sevoflurane anaesthesia and propofol anaesthesia?

Authors calculated PAO2 as FIO2 × (760 – 47) – PaCO2/0.8. (Page 4, Line 131) This equation is valid for low inspired oxygen concentration. In this study, FIO2 was 1.0 (100%) throughout the study period. (Page 3, Line 108-109) In high FIO2, equation for alveolar PO2 (PAO2) should be used as PAO2 = PIO2 {FIO2 × (760 – 47)}– PACO2 × (FIO2 + 1-FIO2/R). Because FIO2 was 1.0 (100%), equation should be PAO2 = FIO2 × (760-47) – PACO2 (PaCO2).

Table 1 and Table 2: Hemoglobin should be Haemoglobin.

Author Response

Hahm and coauthors measured systemic delivery of oxygen (DO2) in patients undergoing thoracic surgery with one-lung ventilation (OLV) and compared differences therein between sevoflurane- and propofol-based anaesthesia. They also evaluated the relationship between arterial oxygen saturation (SaO2) and DO2.

1. have some comments. What are the hypotheses of this study? What did authors want to prove in this study? Please clarify in Introduction.

 : Hypothesis was added in the Introduction and Abstract as

Introduction-“We hypothesised that sevoflurane-based anaesthesia would be associated with a lower DO2 than propofol-based anaesthesia”

Abstract-“Our hypothesis was that propofol-based anaesthesia would show a higher DO2 value than sevoflurane-based anaesthesia.”

2. Measurements of haemoglobin, oxygen saturation, and cardiac output are necessary to calculate DO2. It may be enough to deal with each abnormal value without calculating DO2. Why is it important to calculate DO2?

: We think DO2 gives an information on the net effect. Importance of calculating DO2 is added in the Introduction as “It is thus a more comprehensive (and important) marker of patient status than arterial oxygen saturation (SaO2) [5], being based on haemoglobin, oxygen saturation, and cardiac output.”

3. Why did authors compare sevoflurane and propofol for DO2? Are both anaesthetics predicted to have a different effect on DO2? Please clarify.

 : We assumed sevoflurane would show a lower DO2 than propofol due to its impairment of hypoxic pulmonary vasoconstriction. The following was added in Introduction

“These anaesthetics may show different associations with DO2, based on their differential effects on SaO2 and cardiac output (which are the major determinants of DO2). Inhalation anaesthetics, including sevoflurane, are thought to reduce hypoxic pulmonary vasoconstriction [10,11], thereby increasing the ‘shunting’ of non-oxygenated blood during OLV and thus causing lower SaO2 [12]. Notably, lower SaO2 can decrease DO2. It is not clear whether sevoflurane or propofol yields a higher cardiac output [13-16]. Therefore, in the present study, we measured DO2 in patients undergoing thoracic surgery with OLV and compared differences therein between sevoflurane- and propofol-based anaesthesia. We hypothesised that sevoflurane-based anaesthesia would be associated with a lower DO2 than propofol-based anaesthesia.”

4. Remifentanil was infused at 0.05 μg kg-1 min-1 during surgery. (Page 3, Line 101-102) The dose of remifentanil was very low. It may be difficult to maintain blood pressure and heart rate with such low dose of remifentanil intraoperatively. How did authors deal with intraoperative analgesia? Was epidural anaesthesia performed in this study?

: During thoracic surgery, patient frequently show low blood pressure by surgical manipulation and fluid restriction. Thus, patients usually require low level of remifentanil infusion. Bolus administration of remifentanil (0.3 mg /kg) was used for intubation and during intensive surgical stimulation This was added in the Methods section.

: Bolus remifentanil was not administered during measurement times. Patients enrolled in the study did not receive epidural.

5. How did authors administer propofol? Did authors use target controlled infusion?

 : Regular infusion pump was used. Infusion rate was 80 – 120 mg/kg/min (mean 102 ± 19 mg/kg/min). The following was inserted in the Methods section L103-104

“Propofol was administered using an infusion pump in the range of 80-120 ug/kg/min.”

6. In this study, DO2 increased with time. (Figure 2) Why did it increase with time? Please discuss.

 This increase of cardiac output seems to be the cause of increase of DO2 with time. This was added in the Discussion section (L246) as the following

“The increase in DO2 with time seems to be due to the increase in cardiac output over time.”

7. In this study, plasma lactate level was higher in the Sevoflurane group than in the Propofol group. Why was it higher in the Sevoflurane group?

 : The aetiology is not clear but may be related to a tendency of lower stroke volume in sevoflurane group than propofol group. The following was added in the Discussion

“In this study, the plasma lactate level was higher with sevoflurane than with propofol. The aetiology of this difference is unclear, but may have been related to the tendency towards a lower stroke volume in the sevoflurane group than in the propofol group.”

8. Authors found that SaO2 was not correlated with DO2. Was this phenomenon applicable to sevoflurane anaesthesia and propofol anaesthesia?

: Both sevoflurane and propofol anesthesia showed no significant correlation between SaO2 and DO2. This was inserted in the Result section with correlation coefficient (L210-211).

9.  Authors calculated PAO2 as FIO2 × (760 – 47) – PaCO2/0.8. (Page 4, Line 131) This equation is valid for low inspired oxygen concentration. In this study, FIO2 was 1.0 (100%) throughout the study period. (Page 3, Line 108-109) In high FIO2, equation for alveolar PO2 (PAO2) should be used as PAO2 = PIO2 {FIO2 × (760 – 47)}– PACO2 × (FIO2 + 1-FIO2/R). Because FIO2 was 1.0 (100%), equation should be PAO2 = FIO2 × (760-47) – PACO2 (PaCO2).

: Thank you very much. The formula and the results (Alveolar-arterial O2 difference) (Table 2) were corrected accordingly.

10. Table 1 and Table 2: Hemoglobin should be Haemoglobin.

: Corrected. Thank you.

Reviewer 2 Report

I suggest minor changes:

Results:

I suggest not to write "Stroke volume was higher in the propofol group, but only significantly so .... "Or" Heart rate was higher in the Sevoflurane group than the Propofol group but the difference was not statistically significant ". Since the difference is not significant, it cannot be described as "higher". The authors should only state the absence of difference.

Discussion

I suggest adding in the limitations, the limits of the measurement of cardiac output using the CardioQ. Since DO2 is calculated from cardiac output values, uncertainties in measuring cardiac output may mitigate the results.

I hope this will help.

Author Response

1. I suggest minor changes. Results:

I suggest not to write "Stroke volume was higher in the propofol group, but only significantly so .... "Or" Heart rate was higher in the Sevoflurane group than the Propofol group but the difference was not statistically significant ". Since the difference is not significant, it cannot be described as "higher". The authors should only state the absence of difference.

: Changed accordingly as follows

“Stroke volume was higher in the propofol group. Heart rate and cardiac out were not different between the two groups.”

2. Discussion

I suggest adding in the limitations, the limits of the measurement of cardiac output using the CardioQ. Since DO2 is calculated from cardiac output values, uncertainties in measuring cardiac output may mitigate the results.

: Thank you for your suggestion. We added your point in the limitation section as followings

“Firstly, we used oesophageal Doppler monitoring to measure cardiac output instead of the thermodilution technique, However, a pulmonary artery catheter is rarely used for thoracic surgery, while oesophageal Doppler monitoring has high validity for determining changes and trends in cardiac output and is closely correlated with pulmonary artery catheter and echocardiography data [22-24]. However, DO2 is calculated based on cardiac output values on oesophageal Doppler monitoring, and uncertainties in the cardiac output measurements may have influenced the results.”

3. Moderate English changes required

: Second round of English editing was additionally performed by the following English editing company according to your suggestion.

“English revision: The English in this document has been checked by at least two professional editors, both native speakers of English. For a certificate, please see: http://www.textcheck.com/certificate/G8vRAo”

: Our original paper was previously English-edited by the following English editing company.

English revision: The English in this document has been checked by at least two professional editors, both native speakers of English. For a certificate, please see: http://www.textcheck.com/certificate/Plredo”

Round 2

Reviewer 1 Report

Authors answered all of my questions.

I have no more comments.